# Nursing Care for Metastatic Bone Cancer: Trends for the Future

**DOI:** 10.3390/ijerph20156483

**Published:** 2023-07-31

**Authors:** Debra Penrod, Brandon Hirsch

**Affiliations:** 1Nursing, School of Health Sciences, Southern Illinois University, Carbondale, IL 62901, USA; 2Radiological Sciences, School of Health Sciences, Southern Illinois University, Carbondale, IL 62920, USA; bhirsch110@siu.edu

**Keywords:** radiation dose hypofractionation, palliative care, oncology nursing, pain management, COVID-19

## Abstract

To effectively treat patients and minimize viral exposure, oncology nurses and radiology departments during COVID-19 had to re-examine the ability to offer palliative radiation treatments to people with metastatic bone cancer. Decreasing potential exposure to the virus resulted in extra measures to keep patients and personnel safe. Limiting radiotherapy treatments, social distancing, and limiting caregivers were a few of the ways that oncology patients were impacted by the pandemic. Hypofractionated radiation therapy (HFRT), or the delivery of fewer higher-dose treatments, was a method of providing care but also limiting exposure to infection for immunocompromised patients as well as healthcare staff. As oncology radiation centers measure the impact of patient care during the pandemic, a trend toward HFRT may occur in treating the painful symptoms of bone cancer. In anticipation that HFRT may be increasingly used in patient treatment plans, oncology nurses should consider patient perspectives and outcomes from the pandemic to further determine how to manage future trends in giving personalized care, and supportive care.

## 1. Introduction

The COVID-19 pandemic impacted healthcare in many ways, one of which was the ability of patients to receive treatments for cancer. Studies have shown that people being treated for cancer had a 3.5 times higher risk of severe COVID-19 symptoms compared to other people [1]. A World Health Organization (WHO) early situation report stated that patients particularly at risk included people who were receiving chemotherapy, radical radiotherapy, immunotherapy, antibody treatments, targeted treatments, stem-cell transplants, and immunosuppressant therapy [2]. To effectively treat patients and minimize viral exposure, oncology nurses, as well as other healthcare providers, had to re-examine patients’ needs for optimal outcomes. Strategies were enacted to offset the potentially deadly threat of COVID-19 to cancer patients, which included postponing adjuvant chemotherapy and any elective surgeries, and delivering hypo-fractionated radiotherapy to treat metastatic bone cancer whenever possible [3]. The physical spaces to administer treatments were altered for safe distancing to avoid infection and telemedicine was substituted for face-to-face checkups and consultations. Oncology nurses adapted to the pandemic by not only performing interventions, as per protocol, but also performing other tasks necessary during the pandemic.

The purpose of this paper is to examine how oncology nurses cared for patients during the pandemic and, specifically, to discuss the role of hypofractionation therapy during COVID-19. Preliminary results of hypofractionation therapy during this time are reviewed as well. As researchers examine patient care outcomes and patient perspectives because of actions taken during COVID-19, healthcare providers can make informed decisions to improve care in decreasing anxiety, fear, and pain that often accompany patients with metastatic bone cancer. Lastly, this paper describes a multidisciplinary radiotherapy program with hypofractionation as a future direction in oncology care that could encompass some of the lessons learned from the pandemic.

## 2. Oncology Nursing Care in General during the Pandemic

The challenges that oncology nurses face when aiding patients with cancer intensified further during COVID-19. The primary focus of oncology care, while radiation therapy occurred pre-pandemic, was centered on monitoring patients for immunosuppression, anemia, fatigue, anxiety, and depression [4]. Like other nurses across the healthcare setting during the pandemic, oncology nurses enforced PPE protocols, disinfected treatment areas in-between uses, and spaced treatment areas to accommodate social distancing [2]. In addition to monitoring immunosuppression and anemia, oncology nurses became involved in the surveillance of COVID-19 symptoms during treatments and afterward. Telephone triage was the main method of tracking cancer patients’ symptoms post-treatment, along with referrals to community-based COVID-19 assessment clinics.

Paterson et al. identified the emotional strain placed on oncology nurses to deal with patients’ anxiety and depression during COVID-19 [2]. In some countries, cancer patients were advised to self-isolate for up to six months [5]. Nurses had to find creative ways to meet cancer patients’ needs psychologically, socially, and spiritually using smartphones and electronic devices that allowed patients to connect with family and friends to offset social isolation [2]. During the pandemic, nurses connected with patients electronically to inform them of diagnosis, treatment, and follow-up, as well as survivorship, palliation, and bereavement.

One of the most dynamic ways that oncology nurses impacted cancer patients’ health at home was by instilling self-confidence, so patients believed that they were capable of self-care, and, further to this, being available via phone for problem-solving as special situations arose. From 1 March to 31 October, research from Mumbai, India, illustrated this point, as 85% completed their scheduled treatments with 15% citing fears related to potential COVID-19 infection and inability to commute for treatment [6]. Although the authors stated that this rate was higher than before COVID-19, the fact that 85% of patients completed therapy points to how well nurses supported patients’ needs with information sharing and social support.

Several studies have attempted to measure patients’ perspectives of the changes in oncology nursing interventions during the pandemic. Common themes in retrospect included patients’ understanding of the need to be careful and stay safe coupled with feelings of isolation [7,8]. Patient interviews following COVID-19 also revealed that cancer patients did not feel that they were alone with restrictions on life, because everyone was socially isolated during the pandemic, and patients reported an increased level of resiliency [7].

## 3. Background on Hypofractionation and Use during the Pandemic

Radiation oncologists balance radiotherapy treatments for fighting cancerous lesions with the potential of harm to healthy tissue. One intent of radiation treatments for bone metastasis has been to decrease skeletal-related events (SREs) which can include pathological fractures, immobility, and spinal cord compression [9]. SREs often result in decreased quality of life and extreme pain. Standard fractionation treatments were widely adopted in the humble beginnings of radiation oncology to preserve normal cell function after early casualties suffered radiation injury in a novel medical science.

Technological advances featuring computational applications that design radiation treatments with reconstruction algorithms applied to computed tomography imaging have greatly improved the field of radiation oncology [10]. Improved beam modulation through intensity-modulated radiation therapy (IMRT) and modulated arc therapy have allowed for increased radiation doses to targeted cancer cells, while also lowering doses to normal cells. Figure 1 is an example of a patient with a right femur tumor, who is receiving parallel opposed beams delivering 3000 cGy radiation. These new technologies have allowed radiologists to sculpt doses around the target, while protecting normal structures.

Hypofractionated radiation therapy (HFRT) is the term associated with the delivery of radiation doses with fewer treatment sessions but larger doses per treatment session for cancer treatment. HFRT protocols were utilized as a method not only for infection control during COVID-19, to maintain oncology care, but also to limit exposure to immunocompromised patients as well as healthcare staff. Fewer higher-dose treatments in treating patients with cancer represented further effort, during COVID-19, to provide care while also limiting exposure to infection for immunocompromised patients as well as healthcare staff [11]. During COVID-19, the benefits outweighed the risks, as the patients spent less time in the hospital receiving treatment, which also reduced costs for patient travel. For the healthcare facility, scheduling treatments was less laborious with patients receiving fewer total treatment sessions [12]. Regarding pain control, several clinical trials prior to the pandemic demonstrated little difference in pain control when comparing a variety of dose fractionation schemes for bone metastasis [13]. The disadvantages of HFRT, which include the patient’s potential need for retreatment (7–25%), a higher pathological fracture rate, spinal cord compression, and acute radiotherapy toxicity, were outweighed by the reduction of infection spread among this vulnerable population, as well as among families and healthcare personnel [13,14].

In 2022, Piras, Venuti, D’Aviero, Cusumano, Pergolizzi, Daidone, and Boldrini performed a systematic review, utilizing Preferred Reporting Items for Systematic Reviews and Meta-analyses (PRISMA) guidelines, to study modifications that healthcare providers incorporated to treat all types of cancer in patients during COVID-19, which included HFRT [15,16] Two hundred and eighty-one papers were reviewed, including studies on COVID-19 and radiotherapy, with 28 studies recommending HFRT scheduling as an appropriate treatment during the pandemic [17,18,19,20,21,22,23,24,25,26,27,28,29,30,31,32,33,34,35,36,37,38,39,40,41,42,43]. All the papers reviewed suggested that HFRT use be increased during the pandemic for all cancers. One study, by Piras et al., highlighted that HFRT was a successful treatment option providing satisfactory local control of cancer balanced with the level of toxicity [15,44]. Regarding palliative radiotherapy management, shorter courses were recommended by two sources as well as single fractionation for palliative reasons [13,26,45,46,47,48]. A European Study Group of Bone Metastasis (GEMO) recommended mono-fractionated radiotherapy for painful bone metastasis and spinal cord compression [49].

Piras, et al. concluded, in their systematic review, that most papers found that the use of HFRT reduced times at the hospital, which decreased the potential for infection of all concerned [15,43,50,51]. Furthermore, most researchers found that single-fractionated treatments were preferred for palliative radiotherapy [13,52,53,54]. Data has yet to be collected on the effectiveness of HFRT during the pandemic but a study from the Tata Memorial Center in Mumbai, India, indicated that, from 1 March 2020, through to 31 October 2020, 11,000 new patients were treated for cancer (a decrease of 63% from 2019) [6]. As part of this study, 4256 patient cases were examined for impacts related to COVID-19, 74% of which were treated curatively and 26% palliatively. The most prevalent change during this period was the switch from moderate to ultra-hypofractionated radiotherapy treatments. Eighty-five percent of these patients completed their planned treatments. The 15% non-completion rate was due to testing positive, fear and anxiety of contracting COVID-19, or the inability to commute to the hospital. The authors stated that this data supported the importance of social, psychological, and financial support during radiation treatments. The Radiotherapy Unit of Reggio Emilia (Italy) made the decision early on to avoid postponing any cancer treatments and opted to implement HFRT [55]. A study conducted from this approach between 15 February 2020, to 30 April 2020, revealed a 12% increase in the use of HFRT for patients with bone metastasis for palliative purposes.

## 4. Lessons Learned for Future Nursing Care for Metastatic Bone Cancer

An important lesson to be learned for cancer care, in general, during the pandemic was the need to maintain a connection with patients, preferably in an outpatient setting, or by phone, concerning potential side effects. With the increase in digital follow-up communication and a decrease in outpatient interaction, patients reported a decrease in personal support during the pandemic, which, in turn, promoted more anxiety than usual [7,8]. If hypofractionation treatment for bone cancers increases post-COVID-19, oncology nurses will continue their focus on communicating about the treatment plan, especially regarding pain control, as this continues to be the most prevalent symptom of this cancer and subsequent treatment. Patients who may be treated with hypofractionation include those with primary bone cancers, such as osteosarcoma in children and chondrosarcoma in adults. Primary bone tumors are metastatic and are not currently curable, so the focus of treatment is radiotherapy to increase the quality of life [9].

Pain is the most predominant symptom that bone cancer patients experience for multiple reasons. Between 10% to 30% of patients with bone metastasis experience a pathological fracture, and the most common site for this to occur is the proximal femur [11]. Episodic bone pain results from tumor growth into areas that cause compression of nerves and further pathologic fractures [3]. As cancer progresses with bone destruction, fractures, coupled with nerve compression, cause neuropathic pain. Upon assessment, the patient presents with pain ranging from mild to severe, swelling, muscle atrophy, and spasm.

Another important lesson to be learned from the pandemic is that cancer patients need their caregivers close at hand during consultations and subsequent therapy sessions, if at all possible, to decrease the emotional impact of isolationism with the cancer diagnosis [7,8]. If in-person consultations decrease in frequency with hypofractionation, conference calls that include caregivers are recommended [7]. Anxiety and fear may be expressed by the patient and his or her family, so it is important for nurses to assess the support available, as well as coping mechanisms. Metastatic bone diagnosis may include X-rays, computed tomography, and magnetic resonance imaging. Laboratories need to be monitored, and patients and families need to be informed about aspects such as serum alkaline phosphatase (ALP), calcium, and erythrocyte sedimentation rates (ESR), as well as monitoring for anemia, and leukocytosis [56]. Treatment for non-malignant bone tumors is typically removal, while, for malignant tumors, treatment is directed toward palliation in addressing the source of the pain: debulking or removing the mass through dose fractionation.

## 5. Discussion

Regardless of the changes in radiation dose fractionation, patients still need monitoring for anxiety, fear, and depression, as well as the most predominant symptom of pain. Pain control measures are necessary before, during, and following treatments. Oncology nurses assess the patient’s knowledge and goals for treatment, as well as monitor for symptoms related to bone cancer. The most important part of initial care is empowering the patient and ensuring that the patient and family understand that radiotherapy and subsequent nursing care are aimed at palliation [57]. Emotional support is invaluable to patients facing dose fractionation due to bone cancer.

Pain control is paramount to a patient’s perception of quality of life. Metastatic cancer-induced bone pain (CIBP) is complex with irritation of the nociceptive receptors around the tumor site, as well as neuropathic pain from nerve irritation or damage due to tumor invasion [58]. It is also possible that patients, following radiotherapy treatments, may have an acute pain flare that must be addressed [59]. Due to the complexity of bone pain, a multimodal approach is best. A thorough nursing assessment should be conducted that has the following core elements, as stated by Webb and LeBlanc:Believe the patientTake a careful history, including a detailed pain assessment (location, intensity, quality, onset, radiation, alleviating/exacerbating factors, etc.)Assess the patient’s psychological statePerform a careful examination (including a neurological exam)Obtain diagnostic studies that will help to identify the pain mechanism and review the results yourselfUse a mechanism-based approach to treat the painAssess the response (using formal tools)Individualize the pain treatment approach [60]

Non-pharmacological interventions can help patients as well, which include yoga, psychotherapy, and occupational therapy [61]. Physical activity has been suggested as an effective way to manage pain in advanced bone cancer [62]. Pain control is highly individualized, but the World Health Organization developed guidelines for approaching pain control with four potential steps in the WHO Cancer Pain Ladder (Figure 2).

Figure 2 depicts the four steps of the WHO Revised Pain Ladder, which match levels of pain (mild, mild–moderate, and moderate–severe) with pharmacological approaches and non-pharmacological approaches that include interventional and minimally invasive procedures. Webb and LeBlanc stated that the appropriate opioid dose is the lowest dose that relieves the patient’s pain and maximizes function with the least adverse effects [60]. To minimize pain associated with inflammation from radiotherapy, nurses often administer dexamethasone pretreatment [63]. Corticosteroids address the pain that occurs from initial swelling caused by the tumor itself as it compresses the nerve roots or spinal cord, as well as any swelling that occurs because of cancer treatment [60]. When considering pain control for bone metastasis in active treatment, palliative care, or for end-of-life care, the Centers for Disease Control and Prevention’s guidelines for opioid prescription do not apply [64].

## 6. Future Directions in Cancer Care

As patients continue to undergo single-fraction and hypofractionated radiotherapy treatments, clinics seeking to provide holistic care could provide opportunities for oncology nurses to make an impact on this population. Fairchild, et al., developed a rapid access palliative radiotherapy program (RAPRP) to address the needs of patients who were receiving one radiotherapy treatment, consolidating care in a timely and more efficient manner [65]. Their pilot study involved 71 patients who were served at a single clinical facility. The goal was to maximize the patient’s own time at home rather than multiple individual visits. The multidisciplinary clinic created from the program placed the oncology nurse as the coordinator of care among a nurse practitioner, pharmacist, social worker, registered dietician, and occupational therapist.

Pituskin, et al., revisited the growth of the RAPRPs in 2022 and offered further benefits for patients undergoing single-dose fractionation [63]. The cost of repeat visits falls on the patient with multiple dose fractionation, as well as individual visits to the oncologist, therapist, dietician, pharmacist, and social worker. Some patients from the pilot study stated that money was saved as well as time. Patients reported an average savings of US$152 per visit, due to decreased travel mileage, as well as nearly eight hours of travel time, compared to multiple visits to each specialty involved in care. The registered nurse coordinated multidisciplinary team assessments by performing triage, administering medications, hydration, and ongoing monitoring of patient progress throughout the stay. The nurse also ensured that patients and families were informed of the care plan, as well as changes in status. Following radiotherapy, the registered nurse documented recommendations and performed patient and family education. Follow-up care was performed by the nurse as well, via telephone.

## 7. Conclusions

During the COVID-19 pandemic, oncology nurses, along with radiation oncologists and radiologists, were faced with challenges like many other healthcare providers and divisions, particularly concerning the balance between infection control and indications for interventions and treatment. Actions taken, such as limiting patient exposure by reducing radiotherapy treatments, instituting telemedicine interventions versus live visits and social distancing, were taken to decrease the spread of a global pandemic. By utilizing evidence-based practice from COVID-19, changes to oncological healthcare management may change, which may include the expanded use of telemedicine and hypofractionation therapy. The growth of telemedicine and hypofractionation may allow patients in the future to be treated with fewer visits, which reduces infection transmission and costs. The optimal radiation prescription is a complex decision made by the radiation oncologist.

Regardless of these potential changes to treatment, nursing care to reduce symptoms, such as pain before, during, and following treatment, remains the same. In fact, reducing the number of visits helps patients consolidate visits through a multidisciplinary approach, which saves the patient money, as well as increases their quality of life at home. As healthcare delivery systems consider adopting rapid access radiotherapy programs, this change may not only benefit patients on multiple levels, but also expand the use of oncology nurses in multidisciplinary clinics as coordinators of care.

## Figures and Tables

**Figure 1 ijerph-20-06483-f001:**
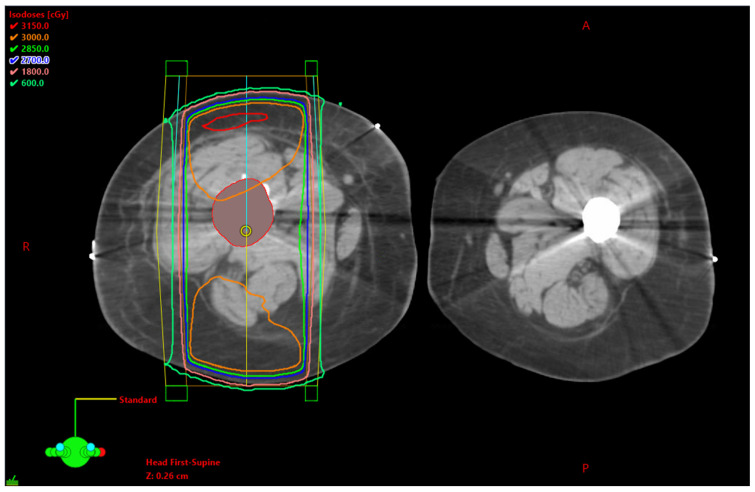
Right femur bone metastasis. The tumor is seen with a red outline. Parallel opposed beams, AP/PA (anterior to posterior/posterior to anterior) arrangement delivering 3000 cGy radiation. (Eclipse TPS Version 13.7, Varian Medical Systems, Palo Alto, CA, USA).

**Figure 2 ijerph-20-06483-f002:**
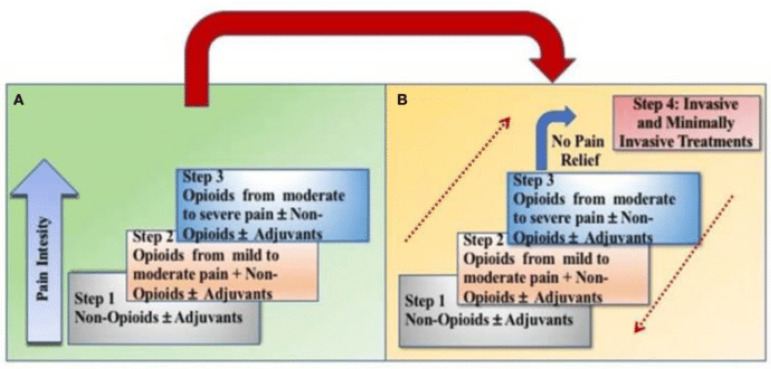
Transition from the original WHO three-step analgesic ladder (**A**) to the revised WHO fourth-step form (**B**). The revised WHO analgesic ladder with an added fourth step for interventional procedures and to show the bidirectional approach to pain control. Contributed by Marco Cascella, MD [61].

## Data Availability

No new data were created or analyzed in this study. Data sharing is not applicable to this article.

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
