# Peer review of "Nursing Care for Metastatic Bone Cancer: Trends for the Future"

_ijerph, 2023, doi:10.3390/ijerph20156483_

Round 1

Reviewer 1 Report

Dear authors, first of all, thank you for your interest in researching this topic.

Regarding the article, there is a lack of coherence between the title and the focus in its development, and the lack of a clear objective only increases the present incoherence. On the other hand, it is advisable to leave primary sources and those contributing to the delimitation of the object and its discussion.

Title: Sure, but not directly related to the abstract.

Abstract: the objective of this article must be made clear, as well as the methodology to be developed. On the other hand, the title and keywords talk about Nursing, but the abstract focuses on another discussion.

MeSH: Not all words are MeSH. It must be reviewed.

Introduction: the delimitation of the object of study should be reviewed since the title speaks of Nursing in the first place. However, this is not mentioned until section number 3. Posing the problematization from radiology or Hypofractionation

Line 105 talks about 28 articles, but only 27 are referenced.

It is peremptory to establish and declare the objective pursued with the development of this article in such a way as to evaluate if the result of this responds to the proposed purpose.

Discussion: the limitations that this has are not presented.

Future directions: again, they are not focused on Nursing.

Conclusions: evaluating is impossible as the objective is not clearly defined.

References: it stands out that there is talk of main actors such as WHO, but reference to secondary sources. Remember to use original quotes

Author Response

Thank you first of all for your review.  I apologize that the manuscript lacked coherence.  I revised the title, and made the abstract more focused on oncology nursing.  Key words were checked to ensure that they were MeSH by a search on https://www.ncbi.nlm.nih.gov/mesh.  

You will find that the entire paper has been reworked to focus more on nursing considerations.  First, challenges for oncology nurses in general is discussed and a few studies reflected on patients' perspectives during that time.  Then, background on hypofractionation and its use during the pandemic is explored as an intervention to decrease the spread of infection.  Next, there is a discussion of lessons learned from a nursing perspective in managing anxiety, fear and pain for the patient with metastatic bone cancer.  Interventions aimed at pain control while also considering personalized care to avoid anxiety and fear are discussed.  Lastly, opportunities for oncology nurses to use these trends to improve patient care through rapid access palliative radiotherapy programs are discussed.   

Line 105:  The original article stated that there were 28 articles but only listed 27 actual articles. 

In regard to WHO as an original source, I cited a reference to the updated pain ladder that added a 4th step to pain control.  This is why I did not cite the original version.

I hope that this manuscript is easier to follow than the previous version.  Sincerely, Debra Penrod

Reviewer 2 Report

Dear authors, thank you for the opportunity to review this manuscript.

In the title, I think that the authors should consider that palliative care is for the person with cancer, not for the cancer.

The authors address the context of Hypofractionated radiation therapy (HRFT) for the treatment of pain due to bone metastasis given the need to minimize the risk of infection during COVID-19. HRFT brought some benefits in terms of logistics for carrying out this therapeutic plan, but without clinical evidence of its effectiveness, with a lower risk of exposure for users and professionals, mainly due to the reduction in the number of visits to the hospital, which also brought financial benefits for some users, as well as services with optimized use of resources.

In session 3, on nursing care, expect more about the role of nurses in managing care for patients with bone metastasis and HRFT. Just as therapeutic plans were readjusted by radiologists in some cases to reduce the risk of infection by COVID-19, with the indication of HRFT, what do palliative care nurses reconsider in their care plans in this context? Although nursing care for pain management before, during, and after radiotherapy is the same, how is this care given? I think this is the central focus, and some different approach must have been considered, no? Since the opportunity for face-to-face meetings was smaller, for example. Also, patients, family members, and nurses were faced with new fears related to the pandemic, as well as the risks arising from HRFT, and case management for individual responses. What has this impacted on this nursing care? I think the epidemiological data that appear in the two paragraphs of this section on bone cancer and pain could be in the introduction. It is important to know about the nurse's role in the management of signs and symptoms from the perspective of palliative care in the face of changes in this therapeutic plan, in relationships, in the organization, in the use of resources, in the environment, all as a result of the pandemic. I miss the description of this perspective based on the author's experience, and I wonder about the motivation for this manuscript of perspectives since it lacks evidence from empirical experience about the performance of the palliative nurse, which could enrich this session, the discussion, and future directions.

In the discussion, the authors present the importance of the multimodal approach to pain and the WHO guidelines for pain control, however, I also highlight the need to broaden the discussions about how necessary the actions performed by palliative nurses in this context were, and instead of detailing pharmacological measures to control cancer pain in this case, the authors could invest, for example, in nursing care, including non-pharmacological estimates, and in the role of nurses navigators.

Author Response

First of all, thank you for your kind review.  

I reworked the entire paper but paid attention to your comments.  The title has been revised to reflect oncology nursing.  I focused on the nurses' response to COVID-19 with the key interventions that were necessary during the pandemic on top of customary duties.  I added in some non-pharmacological interventions for pain control but also focused on decreasing patients' anxiety and fear that intensified during the pandemic.  I tried to focus on the role of the nurse as investigator in monitoring side effects but also for potential of infection.  I cut back on discussing pain control measures as well as that was not the main focus of the manuscript and more about what we can learn from the experience to improve care.  Thank you again!  Debra Penrod 

Reviewer 3 Report

Very Respected Authors,

The main theme of the paper ID: ijerph-2450208 is Hypofractionated radiation therapy (HFRT) as treatment of cancer patients during COVID-19. Authors described the advantages of this method of radiation both for the cancer patients and for the health staff.

Improvement in the section Discussion are needed. The characteristics Webb and LeBlanc which are presented in this section are usually stand in the section Methodology or in the Introduction.  This central part may be modified.

Conclusion is in the agreement with the evidence and arguments presented in the paper. References usually we do not placed in the Conclusion.

The used references are appropriate.

There is only one figure-picture. There is explanation and connection with the text.

Author Response

First of all, thank you for your review.  I reworked the paper in the following manner:

First, challenges for oncology nurses in general is discussed and a few studies reflected on patients' perspectives during that time.  Then, background on hypofractionation and its use during the pandemic is explored as an intervention to decrease the spread of infection.  Next, there is a discussion of lessons learned from a nursing perspective in managing anxiety, fear and pain for the patient with metastatic bone cancer.  Interventions aimed at pain control while also considering personalized care to avoid anxiety and fear are discussed.  Lastly, opportunities for oncology nurses to use these trends to improve patient care through rapid access palliative radiotherapy programs is discussed.

I hope that the discussion section is more complete.  I elected to leave the Webb and LeBlanc citation with the discussion of nursing interventions for pain control in the discussion to show how nurses should continue focusing on individualized care for pain control.  Many of these bulletpoints help address some of the barriers experienced during COVID-19 (lack of connection with healthcare providers, isolationism, and perceived lack of individualized care).

I removed references from the conclusion and added another picture (the revised WHO Analgesic Ladder) which helped me remove the discussion of its use from the paragraph.

Thank you again for your consideration.  Debra Penrod     

Round 2

Reviewer 2 Report

Dear authors, I congratulate you for the answers in the adjustment of the manuscript, however, I noticed that the title remains with a biomedical vision of nursing for cancer and not for the care of the person. It is necessary to revise the list of references, as several others appear in the numerical sequence, such as the number 1. Where the letter p appears in parentheses followed by a number (page 2), is this a direct citation and a reference to the page of the original work? If so, prefer to paraphrase.

Author Response

Hello and thank you again for the review.  These are the changes that were made at your suggestion--

  • The title was changed at the reviewer’s suggestion to reflect patient care rather than practice.  The title now is "Nursing Care for Metastatic Bone Cancer: Trends for the Future"
  • As you review the manuscript, all direct quotes have been removed and areas were paraphrased instead. These are highlighted with comments to delineate the changes.
  • At the reviewer’s suggestion, I matched the reference list to the order in which each reference is used in the manuscript. I found some errors in the references 49-54 which have been fixed.  Some references are used more than once in the manuscript which may explain how the numbers become out of sequence in the manuscript.  For instance, #15 is used 3 times because it was a systematic review of this topic during COVID that was invaluable to this manuscript.  I chose to not highlight these but list them here to avoid confusion.
  • I double-checked our names and affiliation for accuracy.

Thank you again, Debra Penrod